# Unsupervised Domain Adaptation for the Histopathological Cell Segmentation through Self-Ensembling

**Chaoqun Li**                                                 listiralw@gmail.com
*School of Software Engineering, Xi'an Jiaotong University*
**Yitian Zhou**                                                ytzhouuu@gmail.com
*School of Software Engineering, Xi'an Jiaotong University*
**Tangqi Shi**                                            stq5626@stu.xjtu.edu.cn
*School of Software Engineering, Xi'an Jiaotong University*
**Yenan Wu**                                                     wyn@vpx-inc.com
*Frontline Intelligent Technology (Nanjing) Co., Ltd*
**Meng Yang**                                          ym@frontlinemedical.cn
*Frontline Intelligent Technology (Nanjing) Co., Ltd*
**Zhongyu Li**∗                                           zhongyuli@xjtu.edu.cn
*School of Software Engineering, Xi'an Jiaotong University*

**Editor:** TBA

## Abstract

Histopathological images are generally considered as the golden standard for clinical diagnosis and cancer grading. Accurate segmentation of cells/nuclei from histopathological images is a critical step to obtain reliable morphological information for quantitative analysis. However, cell/nuclei segmentation relies heavily on well-annotated datasets, which are extremely labor-intensive, time-consuming, and expensive in practical applications. Meanwhile, one might want to fine-tune pretrained models on certain target datasets. But it is always difficult to collect enough target training images for proper fine-tuning. Therefore, there is a need for methods that can transfer learned information from one domain to another without additional target annotations. In this paper, we propose a novel framework for cell segmentation on the unlabeled images through the unsupervised domain adaptation with self-ensembling. It is achieved by applying generative adversarial networks (GANs) for the unsupervised domain adaptation of cell segmentation crossing different tissues. Images in the source and target domain can be differentiated through the learned discriminator. Meanwhile, we present a self-ensembling model to consider the source and the target domain together as a semi-supervised segmentation task to reduce the differences of outputs. Additionally, we introduce conditional random field (CRF) as post-processing to preserve the local consistency on the outputs. We validate our framework with unsupervised domain adaptation on three public cell segmentation datasets captured from different types of tissues, which achieved superior performance in comparison with state-of-the-art.

**Keywords:** Unsupervised Domain Adaptation, Cell Segmentation, Self-Ensembling

---

∗. Corresponding Author. This work is partially supported by the China Postdoctoral Science Foundation (No. 2019M663670) and the Natural Science Basic Research Program of Shaanxi (No. 2020JQ-030).

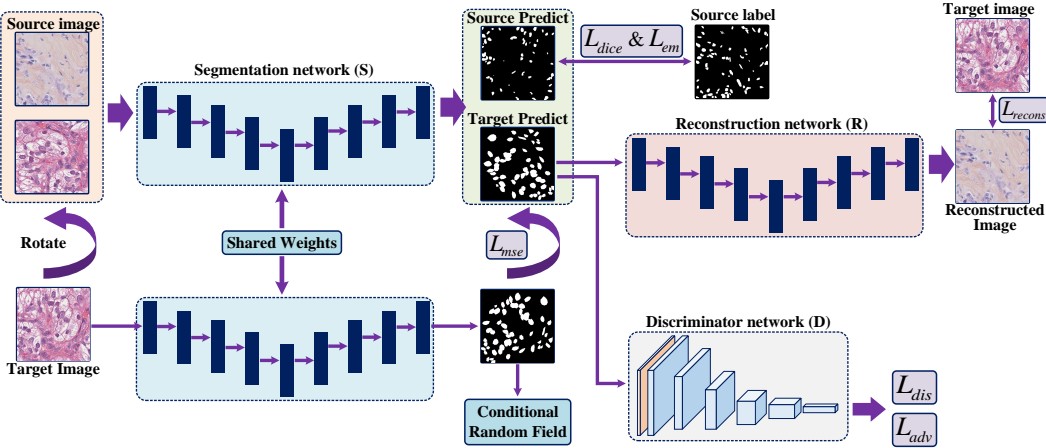

Figure 1: The pipeline of our proposed UDA method for histopathological cell segmentation.

## 1. Introduction

The field of computer-aided digital pathology plays an important role in diagnosis and treatment. In pathological images, it is self-evident that the importance of cell is almost second to none as it is the basic component of creatures. As it always takes doctors a large amount of time, labor, and fund to do segmentation manually, a critical need for more accurate, robust, and low-cost nuclei segmentation methods come into being.

Nowadays, cell segmentation in histopathology images has been extensively studied with a variety of deep learning methods. Inspired by the advance of Fully Convolutional Networks (FCN) (Long and et al., 2015), there is a variety of deep learning methods proposed in the segmentation field like U-Net (Ronneberger and et al., 2015) and DeepLab (Chen and et al., 2017). For example, Chen and et al. (2016) proposed a deep contour-aware network (DCAN) to establish better segmentation by a multi-task learning framework that learns not only probability maps but also clear contours. However, well-annotated datasets for cell segmentation are extremely limited in clinical diagnosis. And unsupervised domain adaptation comes to researcher's vision as it can reduce manual annotation works and tackle generalization problems among data collected from different clinical sites or different modalities. Liu and et al. (2020) proposed a method based on Cycada (Hoffman and et al., 2018) which adds a task re-weighting mechanism along with a nuclei inpainting mechanism to make the framework perform better on data from different organs. Haq and et al. (2020) proposed a framework based on GAN (Goodfellow and et al., 2014) along with a reconstruction network to do segmentation on unlabeled data from different organs.

Despite the above methods have already taken cross domain problems into consideration, there are still multiple challenges in this field. Firstly, current methods may lack robustness and stable performance when segmenting cross-domain cells. This limitation is critical for cell segmentation tasks, where the annotations are hard to obtain. Secondly, existing works only consider the domain adaptation within the same tissue, which cannot be widely applicable in clinical diagnosis.

Taking the challenges above into account, this paper proposes a novel framework for cross-domain cell segmentation. Particularly, as presented in Fig. 1, we first apply a semi-supervised method with self-ensembling. The framework can extract features from the source domain with data augmentation to improve the segmentation performance. Then the unsupervised domain adaptation can effectively transfer cell segmentation results across different tissues or segmentation using different staining methods. The contribution of this paper can be summarized in three aspects: 1) we develop a new framework that contains semi-supervised segmentation with domain adaptation on the target domain; 2) we introduce a semi-supervised framework to do data augmentation in the feature extraction part and a CRF (Boykov and et al., 2004) module in the post-processing part to improve the robustness of the cell segmentation; 3) extensive and comprehensive experiments are carried out on three datasets from different tissues to demonstrate the effectiveness of our proposed methods.

## 2. Methodology

**Semi-supervised Segmentation:** Formally, in the cell segmentation problem, the histopathological image patches from the datasets are input X of size $H \times W \times 3$. Then, we want to predict the segmentation output $\widehat{Y}$ of size $H \times W \times 1$. In the source domain, we also have binary masks with pixel-wise ground-truth label Y of size $H \times W \times 1$ in our framework.

The segmentation network takes images X as input and uses the segmentation predictions $\widehat{Y}$ of the same width and height as output, i.e., $\widehat{Y} = S(X)$. We train S to generate the predictions $\widehat{Y}_s$ through a semi-supervised segmentation method while using the $Y_s$ as the ground-truth input of the source domain. As for the target domain, there is no label for segmentation in the unsupervised domain adaptation problem. And the GAN method sometimes may not be able to obtain enough information from the features in the segmentation network. Therefore, we use the entropy minimization loss to control the weight of the labeled examples, increase the confidence of the segmentation output, and make the model more stable. In practice, the dice-coefficient loss and the entropy minimization loss are more effective than the normally used binary cross-entropy loss. So we use both the dice-coefficient loss and the entropy minimization loss as our segmentation loss:

$$L_{\text{dice}} = 1 - \frac{2.Y_s'.\widehat{Y}_s'}{Y_s' + \widehat{Y}_s'} \tag{1}$$

$$L_{em} = -\frac{1}{H*W} \sum_{h=0}^{H} \sum_{w=0}^{W} \widehat{Y}_s \log(\widehat{Y}_s) \tag{2}$$

where $Y_s'$ and $\widehat{Y}_s'$ are flatten $Y_s$ and $\widehat{Y}_s$ respectively.

In practice, the images from the source and the target domain may differ a lot in staining result, clarity, and direction. Therefore, the data augmentation and the robustness should be taken into consideration. Accordingly, we apply a self-ensembling method by using rotate transformation in a generalized form. And we optimize the consistency loss with a teacher model, which shares its weights with the student model. The key point in the teacher-student learning-based semi-supervised segmentation network is the smoothness assumption. For example, data points close to each other in the image space are more likely to be close in the label space. To be specific, the semi-supervised segmentation tasks can

be learned by optimizing a mean square error loss:

$$L_{mse} = ||\widehat{Y}_{s,\pi} - \widehat{Y}_s||^2 \tag{3}$$

where $\widehat{Y}_{s,\pi}$ is the prediction of the source image which goes through the transformation-consistent regularization named $\pi$.

In above, the target data goes through the model twice to get two predictions under different perturbations. The model assumes a dual role as a teacher and as a student. As a student, it learns a fully-supervised segmentation network. As a teacher, it generates targets to be used by itself as a student for learning through optimizing the mean square error loss. By doing so, the self-ensembling method is applied to the segmentation model to make the performance more stable, which may work better in the diagnosis.

The overall segmentation loss function is then defined as:

$$L_{seg} = L_{em} + \lambda L_{mse} + L_{dice} \tag{4}$$

where $L_{em}$, $L_{dice}$ and $L_{mse}$ are supervised term (the former two) and regularization term. As $L_{em}$ and $L_{dice}$ have similar importance, we set the weights of them at 1. As for $L_{mse}$, we use a time-dependent warming up function $\lambda$ as a weighting factor. This weighting function is a Gaussian ramp-up curve that slowly drop down as:

$$\lambda = k * e^{||e-E||} \tag{5}$$

where E denotes the training epoch, k scales the maximum value of the weighting function, and e defines the peak. In our experiments, we empirically set k to 1.0 and e to 30.

Training S with the annotated source data teaches S to make accurate predictions. At this stage, the segmentation network may not generate correct predictions for target images as there are discrepancies between the source and the target. Therefore, the S needs to generate target domain predictions closer to the source domain predictions by making the distribution of target predictions $\widehat{Y}_t$ closer to $\widehat{Y}_s$. We define the adversarial loss as:

$$L_{adv}(X_t) = -\frac{1}{H' \times W'} \sum_{h',w'} \log(D(\widehat{Y}_t)) \tag{6}$$

where $\widehat{Y}_t = S(X_t)$, and $H'$ and $W'$ are the height and width of discriminator output $D(\widehat{Y}_t)$. This adversarial loss helps S to fool the discriminator so that it could consider $\widehat{Y}_t$ as a source domain segmentation prediction.

This semi-supervised network can be treated as the generator of a GAN (Goodfellow and et al., 2014). And to make the predictions closer to each other, we also need a discriminator. **Discriminator:** We introduce a discriminator D into the framework. It can take the predictions as its input and distinguish whether the input comes from the source domain or the target domain. To train D, we use a cross-entropy loss as:

$$L_{dis}(\widehat{Y}) = -\frac{1}{H' \times W'} \sum_{h',w'} z \cdot \log(D(\widehat{Y})) + (1-z) \cdot \log(D(\widehat{Y})) \tag{7}$$

where z=0 when D takes target domain prediction as input, and z=1 when input comes from source domain.

Moreover, it is possible that these target predictions are not well-correlated with the target input image. A network for reconstructing images from the predictions to a similar appearance as input can ensure the correlation between the input and the prediction.

**Reconstructor:** We use a Reconstruction network R in our framework and consider the segmentation network S as an encoder and the Reconstruction network R as a decoder to reconstruct original images from predictions. It takes the predictions $\widehat{Y}_t$ as inputs and produce the reconstructed image as the output $R(\widehat{Y}_t)$. We calculate the reconstruction loss as:

$$L_{recons}(X_t) = \frac{1}{H \times W \times C} \sum_{h,w,c} (X_t - R(\widehat{Y}_t)^2) \tag{8}$$

where $R(\widehat{Y}_t)$ is the output of the Reconstructor for input $\widehat{Y}_t$, and H, W, C are the height, width, and number of channels of the input image $X_t$.

Overall, we optimize the following total loss when training our framework:

$$L(X_S, X_t) = L_{seg}(X_S) + \lambda_{adv}L_{adv}(X_t) + \lambda_{recons} L_{recons}(X_t) + L_{dis}(\widehat{Y}) \tag{9}$$

where the $\lambda_{adv}$ and $\lambda_{recons}$ are the weights to balance above losses.

At this point, our results have learned information from both the labeled source domain and the unlabeled target domain. Because neighbouring voxels share substantial spatial context, the segmentation results produced by the CNN are smooth. However, local minimization training and noise may still result in some spurious outputs, like small isolated regions or holes in the predictions, which result from the lack of regional spatial information. Accordingly, we employ a fully connected conditional random field (Krähenbühl and et al., 2011) as a post-processing step to achieve more structured predictions and constrain the spatial consistency of the results. For an input image X and its segmentation prediction $\widehat{Y}$, the Gibbs energy in the CRF model is given by:

$$E\left(\widehat{Y}\right) = \sum_i \psi_u\left(\widehat{Y}_i\right) + \sum_{ij,i \neq j} \psi_p\left(\widehat{Y}_i, \widehat{Y}_j\right) \tag{10}$$

$$E\left(\widehat{Y}\right) = \sum_i \psi_u\left(\widehat{Y}_i\right) + \sum_{ij,i \neq j} \psi_p\left(\widehat{Y}_i, \widehat{Y}_j\right) \tag{11}$$

the unary potential is the negative log-likelihood $\psi_u\left(\widehat{Y}_i\right) = -logP(\widehat{Y}_i|X)$, where $P(\widehat{Y}_i|X)$ is the model's output for pixel i. The pairwise potentials in our model have the form:

$$\psi_p\left(\widehat{Y}_i, \widehat{Y}_j\right) = \mu\left(\widehat{Y}_i, \widehat{Y}_j\right) \sum_{m=1}^K w^{(m)}k^{(m)}\left(f_i, f_j\right) \tag{12}$$

where $k^{(m)}$ is Gaussain kernel $k^{(m)}(f_i, f_j) = exp(-\frac{1}{2}(f_i - f_j)^T\Lambda^{(m)}(f_i - f_j))$. The vectors $f_i$ and $f_j$ are feature vectors for pixels i and j in an arbitrary feature space, $w^{(m)}$ are linear combination weights, $\mu$ is a label compatibility function. Each kernel $k^{(m)}$ is characterized by a symmetric, positive-definite precision matrix $\Lambda^{(m)}$, which defines its shape.

The contrast-sensitive two-kernel potentials are used for our image segmentation and labeling, consisting of the appearance kernel and the smoothness kernel. They are defined

Table 1: Segmentation results of four compared methods and our framework.

| Source Domain | TNBC | | | | TCIA | | | |
|---|---|---|---|---|---|---|---|---|
| Target Domain | KIRC | | TCIA | | KIRC | | TNBC | |
| | Iou | Dice | Iou | Dice | Iou | Dice | Iou | Dice |
| DA_ADV (Dong and et al., 2018) | 0.3276 | 0.4911 | 0.4480 | 0.6082 | 0.2914 | 0.4484 | 0.4078 | 0.5712 |
| CBST (Zou and et al., 2018) | 0.3031 | 0.4627 | 0.4015 | 0.5638 | 0.2853 | 0.4413 | 0.3901 | 0.5547 |
| CellSegUDA (Haq and et al., 2020) | 0.5487 | 0.7039 | 0.5802 | 0.7188 | 0.5539 | 0.7067 | 0.4109 | 0.6078 |
| Our framework$_{withoutCRF}$ | 0.551 | 0.7023 | 0.5689 | 0.7233 | 0.5566 | 0.718 | 0.4023 | 0.6051 |
| Our framework$_{withoutEMloss}$ | 0.5523 | 0.7025 | 0.5661 | 0.7211 | 0.5665 | 0.7251 | 0.4271 | 0.6295 |
| Our framework$_{withoutself-ensembling}$ | 0.5607 | 0.7189 | 0.5791 | 0.7301 | 0.5653 | 0.7233 | 0.4308 | 0.6246 |
| Our framework | **0.5683** | **0.7234** | **0.6049** | **0.7413** | **0.5737** | **0.7261** | **0.5263** | **0.6796** |

in terms of the color vectors $I_i$ and $I_j$ and positions $p_i$ and $p_j$ :

$$k(f_i, f_j) = w^{(1)}exp(-\frac{|p_i - p_j|^2}{2\Theta_\alpha^2} - \frac{|I_i - I_j|^2}{2\Theta_\beta^2}) + w^{(2)}exp(-\frac{|p_i - p_j|^2}{2\Theta_\gamma^2}) \qquad (13)$$

the appearance kernel means that nearby pixels which have similar color are likely to be in the same class. The degrees of nearness and similarity are controlled by parameters $\Theta_\alpha$ and $\Theta_\beta$. The smoothness kernel removes small regions that isolated with big ones. Finally, the weights $w^{(1)}$ and $w^{(2)}$ define the relative strength of the two factors.

## 3. Experiment

**Datasets:** There are three datasets used in our experiments. Irshad and et al. (2014) released the KIRC dataset which consists of 463 images of 400×400 pixel size from Kidney Renal Clear cell carcinoma (KIRC). Naylor and et al. (2018) released the TNBC dataset that consists of 50 images of 512×512 pixel size from Triple Negative Breast Cancer Cell (TNBC). Hou and et al. (2020) released the TCIA dataset that consists of 1356 images of 256×256 pixel size from 14 different cancer types. We use 97 images from Stomach adenocarcinoma (STAD) among the dataset to maintain the images from one dataset come from one tissue. All the datasets are extracted at 40× magnification and come from whole slide images (WSI).

**Settings:** We use U-Net (Ronneberger and et al., 2015) as our backbone. We use 80% of the images for training, 10% for validation and 10% for evaluation. During the training, we employ Adam optimizer (Kingma and et al., 2014) to optimize the losses with learning rates of 0.0001, 0.001, and 0.001 used in the segmentation network, discriminator, and reconstructor respectively. We use 0.001 and 0.01 as $\lambda_{adv}$ and $\lambda_{recons}$ respectively and train in total 500 epochs. All experiments are carried out by using Pytorch on a Linux system with 2 RTX 2080Ti and take about 14 GB memory of the graphic cards for 12 hours.

**Results:** In our experiment, we use TNBC (Naylor and et al., 2018) and TCIA (Hou and et al., 2020) datasets as the source domain respectively and the other two datasets as the target domain respectively. Besides, our proposed method is compared with 3 recently proposed related methods. The first one is DA-ADV, a UDA method based on the GAN method which also uses a discriminator like ours and proposed by Dong and et al. (2018). The second one is CBST, another UDA method proposed by Zou and et al. (2018). This is a popular classbalanced self-training framework by generating pseudo labels, which is a different method to transfer the domain from ours. The third one is CellSegUDA, an

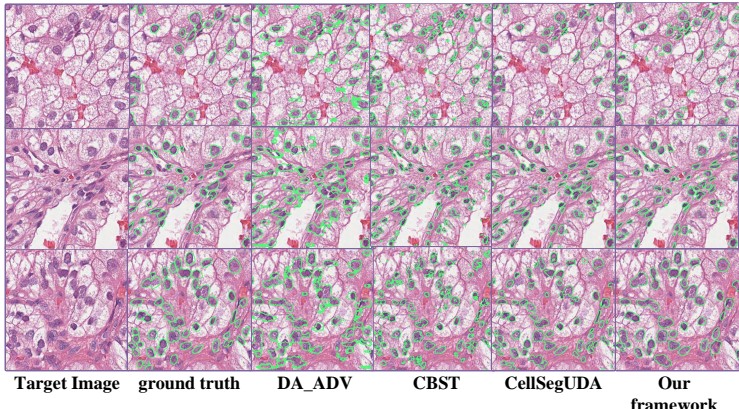

(a) Visualization results randomly chosen from the target dataset.

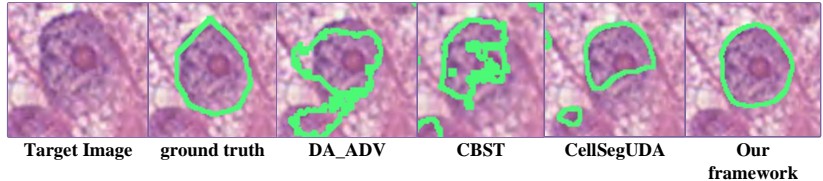

(b) Enlarged visualization results of a single cell.

Figure 2: The visualization results of target domain for segmentation from TNBC to KIRC.

unsupervised adversarial domain adaptation method proposed by Haq and et al. (2020). This method is based on the GAN method and achieved excellent performance on cell segmentation tasks. We also perform ablation experiments to validate the effectiveness of each component in our framework.

To evaluate the segmentation accuracy of the nuclei instances, we use both the Dice and IoU metrics as our validation metrics. As shown in Table 1, compared with the former three baseline methods, our framework's performance has a significant improvement under both the IOU and the Dice metrics. Since the self-ensembling method can enhance the robustness and the GAN network can benefit from it, our framework works better in comparison with the other three methods. And in ablation experiments, with individual components removed, the performance of the framework falls, which proves that every component could contribute to the performance of the framework. And Figure 2 shows the visualization of segmentation results of our framework and 3 compared methods, which indicates that there are significant improvements in our framework. In the results of the target domain, our framework works better than other methods and get a more accurate segmentation result, which benefits from the self-ensembling method and the CRF post-processing step.

## 4. Conclusions

In this paper, we propose a novel unsupervised domain adaptation method for the cell segmentation across different tissues. The method is based on the GAN framework and suitable for the segmentation of whole-slide images from different tissues. We improve the

segmentation performance by introducing a teacher-student learning-based semi-supervised segmentation network, which could help with the data augmentation and to overcome the problem of low robustness. Meanwhile, we use the conditional random field method as our post-processing step to achieve more structured predictions and constrain the spatial consistency of the results. Based on this work, we may apply self-supervised methods in this field and study how to make the domain adaptation more accurate in the future.

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
