# OpenReview forum: "Unsupervised Domain Adaptation for the Histopathological Cell Segmentation through Self-Ensembling"
_MICCAI.org/2021/Workshop/COMPAY — COMPAY 2021_

### Official Review · Reviewer_hpHp · 2021-07-30
**This paper present incremental contributions for solving unsupervised cross-domain segmentation from histopathological images. The ideas of the paper are not new but their combination makes it sound. My main concern about this paper are regarding the clarity of the paper and the quality of the conducted related work studies.  Moreover, I believe a better organization can be structured. Please refer to the cons comments for some constructive feedback.**

**Rating:** 6
**Confidence:** 5

**Review:**


# Recap:
This paper address the problem of unsupervised cross-domain cells/nuclei segmentation from histopathological images, where the idea is to transfer the segmentation model learned from labeled source dataset to target dataset. The proposed method is based on the GAN framework where images from both domain are differentiated through a discriminator and the reconstruction network reconstructs input images from segmented binary masks which act as generator. Additionally, the authors presented a semi-supervised segmentation network via self-ensembling that consider the labeled source domain dataset and transformed target dataset to reduce the differences of outputs that are presented  in staining result, clarity, and direction. Hence achieving better performance on different pathological source and target images. Finally, to preserve the local consistency on the target outputs, the authors introduce conditional random field  as post-processing stage.



# Pros:
1. The authors paid special attention on formulating their method. For instance, since GANs are prune to model collapse on small dataset, the authors relied on entropy minimization loss to control the weight of the labeled examples while stabilizing the model performance.

2. The authors replaced typical binary cross-entropy loss used for full segmentation supervision by the combinations of dice-coefficient loss and the entropy minimization loss.

3. To improve segmentation performance on the target data, data augmentation embedded with self-ensembling method by using rotate transformation is devised to improve generalization form.

4. The authors presented a time-dependent warming up function λ as a  weighting factor for supervised loss and regularization loss based on  Gaussian ramp-up curve.

# Cons:

## Introduction:

1. Paragraph 1: the motive and the advantages behind devising an automatic segmentation method are well known and clear to the community. Therefore, the authors are welcomed to reduce this paragraph and are recommended to focus more on the problem statement of unsupervised domain adaptation.


2. Paragraph 2:
* the motive behind moving to UDA is illustrated in this section from one point of view that is to reduce manual annotation efforts. However, UDA is larger than this! it tackle generalization among data collected from different clinical sites, or data that contains domain shift problem, for instance images coming from different clinical sites, or different modalities, etc. Therefore, the authors are recommended here to clearly indicate how the term “domain adaptation” is suited for this study. Here, the authors might speak about the type of the data that are going to be used for supervising the segmentation method and where it will be applied on the unsupervised target data.

* Within the same paragraph, state of the art is rushed! There are a wide range of literature on Unsupervised Domain Adaptation that should be cited and discussed. Moreover, the discussed state of the art is shallow. The differences between the authors method and state of the art methods should be discussed. Here are some of the recent methods that could be covered:

1. OLVA: Optimal Latent Vector Alignment for Unsupervised Domain Adaptation in Medical Image Segmentation.
2. Synergistic image and feature adaptation: Towards cross-modality domain adaptation for medical image segmentation.
3. Unsupervised bidirectional crossmodality adaptation via deeply synergistic image and feature alignment for medical image segmentation.
4. Data efficient unsupervised domain adaptation for cross-modality image segmentation.

* Figure 1: It is not very clear what is the type of images that are associated with the source domain!

* Page 4, typo: spaceLaine

## Section Experiment:
* Please add a figure that shows the images coming from the different datasets used in this paper. And if possible, it is recommended to add some analysis on the statistical differences between them. In this way, we can understand where the domain shift lies.

* Table 1. should be described deeply and not only mentioning that the method performed better. For instance, the effect of which is the source domain dataset and which is the target dataset and their performance should be described and illustrated.

* Why 97 images among the TCIA dataset are used instead of 1356 images !

## Section Results:
* please clearly indicate that the target data are TNBC to KIRC to remind the reader.

* Since the dice is used as a loss function, only mIoU would be informative measure. Here it would be also efficient to report HDD, Precision and Recall to report the sensitivity of the method.

* The authors mentioned “ our framework works well in a blurry image and get a larger and more accurate segmentation result, ” –> How to quantify that the segmentation is good on blurry images? any metric over them? Any visualization and special experimental setup?

## Conclusions:
* it is recommend to state how other domains can benefit from this model rather than recapping what has been already said in the paper. Discussing some improvements or perspective would be beneficial.

---

### Official Review · Reviewer_pR1o · 2021-08-14
**Unsupervised domain adaptation through self-ensembling**

**Rating:** 7
**Confidence:** 4

**Review:**

This paper presents an unsupervised domain adaptation method for histopathology cell segmentation through self-ensembling. Several strategies have been employed in the manuscript including self-ensembling in semi-supervised setting, generative adversarial training for unsupervised domain adaptation, and conditional random field for post-processing. Overall, this paper is well written and experimental results on several datasets demonstrate good performance. I have following suggestions to further improve the quality of manuscript:
1.	As mentioned before, several strategies have been put together and demonstrate good experimental results. The new technological contribution should be justified and elaborated clearly. For example, either self-ensembling or conditional random field has been explored in the medical image setting. Below are just a few examples for reference.
Zhang, Yue, et al. "Task driven generative modeling for unsupervised domain adaptation: Application to x-ray image segmentation." International Conference on Medical Image Computing and Computer-Assisted Intervention. Springer, Cham, 2018.
Li, Xiaomeng, et al. "Transformation-consistent self-ensembling model for semisupervised medical image segmentation." IEEE Transactions on Neural Networks and Learning Systems 32.2 (2020): 523-534.
Fu, Huazhu, et al. "Deepvessel: Retinal vessel segmentation via deep learning and conditional random field." International conference on medical image computing and computer-assisted intervention. Springer, Cham, 2016.
Dou, Qi, et al. "3D deeply supervised network for automated segmentation of volumetric medical images." Medical image analysis 41 (2017): 40-54.
2.	In equation (3), what are the balance weights for different terms and how are they determined? Similar to Eq. (9).
3.	In Table 1, I appreciate the authors’ efforts on conducting extensive studies to validate the feasibility of the proposed method. For the ablation studies, I wonder if it is possible we can add modules one by one. Thus, we can know how much contribution each module has made.
4.	In Figure 2, to better differentiate the difference of different methods, I suggest authors can present some magnified regions to highlight the outputs.

---

### Decision · Program_Chairs · 2021-08-25

Accept